# The Potential Role of Gut Bacteriome Dysbiosis as a Leading Cause of Periprosthetic Infection: A Comprehensive Literature Review

**DOI:** 10.3390/microorganisms11071778

**Published:** 2023-07-09

**Authors:** Konstantinos Valtetsiotis, Alberto Di Martino, Matteo Brunello, Leonardo Tassinari, Claudio D’Agostino, Francesco Traina, Cesare Faldini

**Affiliations:** 1Department of Biomedical and Neuromotor Science-DIBINEM, University of Bologna, 40127 Bologna, Italy; konstantinos.valtetsiotis@ior.it (K.V.); matteo.brunello@ior.it (M.B.); leonardo.tassinari@ior.it (L.T.); francesco.traina@ior.it (F.T.); cesare.faldini@ior.it (C.F.); 21st Orthopedic and Traumatology Department, IRCCS Istituto Ortopedico Rizzoli, 40136 Bologna, Italy; 3Orthopedics-Traumatology and Prosthetic Surgery and Hip and Knee Revision, IRCCS Istituto Ortopedico Rizzoli, 40136 Bologna, Italy

**Keywords:** periprosthetic joint infection, total hip arthroplasty, total knee arthroplasty, joint, gut, microbiota, dysbiosis, infection, IBD, Crohn’s disease

## Abstract

(1) Background: Periprosthetic joint infections (PJIs) represent a small yet important risk when undertaking a joint arthroplasty; they occur in approximately 1–2% of treatments. These infections create a medical and financial burden for patients and healthcare systems. Despite the introduction of recognized best clinical practices during arthroplasty operations, it is not yet possible to further reduce the risk of infection after surgery. The purpose of this review is to raise awareness of the potential role of gut dysbiosis in the development of PJIs and to highlight the potential of the gut bacteriome as a possible target for preventing them. (2) Methods: We compiled all the available data from five databases, examining the effects of gut dysbiosis in human and murine studies, following PRISMA guidelines, for a total of five reviewed studies. (3) Results: One human and one murine study found the Trojan horse theory applicable. Additionally, inflammatory bowel diseases, gut permeability, and oral antibiotic ingestion all appeared to play a role in promoting gut dysbiosis to cause PJIs, according to the other three studies. (4) Conclusions: Gut dysbiosis is linked to an increased risk of PJI.

## 1. Introduction

Total joint arthroplasty (TJA) is a relatively safe surgical procedure; however, it carries a risk of periprosthetic joint infections (PJI) for total hip and knee replacements, at around 1% [1]. The majority of PJIs occur within the first 2 years after surgery and there is a lower risk of development in the 2–10-year post-operative window [2,3]. Diagnostic criteria for PJI include multiple prerequisites that vary according to the releasing scientific society. The International Consensus Meeting defined major and minor criteria for the diagnosis of PJI, with major criteria being: (1) Two identical organisms are found from two separate cultures from the periprosthetic region, (2) the presence of a sinus tract communicating with the joint, and (3) at least three minor criteria from a list (includes several tests for elevated leukocyte levels, positive histological analysis of periprosthetic tissue, and elevated inflammation biomarkers) [4,5]. PJI of the hip or knee can be monomicrobial or polymicrobial, with approximately 70% of cases being monomicrobial, while the other 25% are polymicrobial; in the remaining 5%, PJIs are culture negative, meaning that a pathogen could not be identified [6]. The most common bacteria involved in TJA infections are aerobic Gram-positive (82%), of which the coagulase-negative staphylococcus species accounts for 34% of total infections, making it the most prominent species. It is followed by anaerobic bacteria (13%), Gram-negative aerobic bacteria (11%), fungi (3%), and mycobacteria (0.5%) [6].

The treatment of PJI involves a combination of surgical intervention and antibiotic therapy. Therefore, revision TJAs due to PJI create an additional burden to the healthcare system. Surgical therapy is required to eradicate the infection, as the formation of biofilm prohibits the treatment of antibiotic-only therapy. In acute infections, it is possible to perform a Debridement and Implant Retention (DAIR) procedure. It involves a thorough debridement (cleaning and removal of infected tissue) along with the exchange of modular components, followed by antibiotic treatment. If the infection is chronic, the revision surgery can be performed in one or two stages. In the former, it involves the removal of the infected implant and replacement with a new one in a single operation, along with a thorough debridement and irrigation of the surrounding tissues. In the Two-Stage Exchange, the infected prosthesis is removed during the first surgery, extensive debridement is performed, and a temporary antibiotic spacer is implanted to keep the joint space. The patient then undergoes a period of systemic antibiotic treatment to clear the residual infection. In the second surgical stage, the spacer is removed, and a new prosthesis is implanted [7]. Although PJI rates are similar, the two-step procedure offers more control over the duration of antibiotic therapy and allows for intermediate adjunctive surgical debridement procedures if the infection is persistent [8]; this flexibility usually makes the two-step procedure the preferred choice of surgeons. Patients undergoing two-stage revision surgery for primary joint arthroplasty require longer inpatient treatment and more outpatient visits, and they are also more likely to be readmitted because of complications. Moreover, the entire surgery-related procedure is more expensive when compared to primary TJA [9]. In the United States, it is estimated that by 2020 the annual cost for revision surgeries due to PJI averaged USD 1.62 billion [10], and by 2030 total hip and knee replacements will alone amass to USD 1.85 billion [11]. Europe follows a similar trend, with primary TJAs (and by extension revision surgeries) expected to rise in the coming years as the population continues to age [12,13,14]. 

Periprosthetic infections can be classified in two categories of origin: (1) intraoperative, which are from microbes introduced exogenously during operation, and (2) haematogenous, which are from microbes introduced endogenously through the bloodstream. Approximately one-third of infections that occur are from the latter category [15,16], which includes any potential gut microbiota-sourced infection. The GI tract hosts between 1013 and 1014 microbial cells, which is approximately equal to the number of human cells that make up the body [17]. The gut microbiota is composed of a diverse range of organisms, including bacteria, archaea, fungi, protists, and algae [18] There is large interpersonal variability in the composition of the gut microbiota. However, while the ratio may vary, bacteria are by far the most prominent microbe. There are five major phyla that account for the majority of bacteria found in the gut: firmicutes, proteobacteria, bacteroides, actinobacteria, and fusobacteria [19]. Gut microbiota colonisation begins after birth, starting with the first breastfeeding. However, given that the uterus is not perfectly sterile, it is likely that the first colonisation of organisms in the gut microbiota may begin in utero [20]. The gut microbiota keeps developing throughout the early years of life, becoming more diverse overtime. Though it was believed that the gut microbiota reached a point close to maturity by 3 years of age, more recent analysis suggests that the maturation process may continue well into late childhood [20]. Factors affecting the development of the gut microbiota include but are not limited to: the consumption of prebiotics and probiotics, the administration of antibiotics, non-antibiotic medications, breast milk or formula feeding, the delivery method, genetics, and diet [21,22]. Bacteriophages and the gut virome they create are also considered to have an important role in modulating the gut microbiota [23]. Elderly adults (>65 years old) experience a shift in composition, with a decrease in bacteroides and firmicutes, which in those of a younger age tend to be the two most prominent phyla. Additionally, elderly adults appear to have a significantly higher interindividual variability compared to younger individuals [24]. Long-term GI microbiota composition is also determined by the overall diet of an individual [20]. Gut microbiota was considered to have a commensal relationship with humans (hence the existing bacteria were referred to as commensals) [25]. However, under healthy GI conditions, the relationship is likely to be more mutualistic [25,26] because microbiota provide various benefits to the host including drug metabolism, protection against harmful bacteria, regulation of the immune and inflammatory response, and maintenance of the GI barrier integrity [26].

In a normal gut environment, gut microbiota and the human body reach a homeostatic balance where the quantity and quality of microbiota are regulated, either independently or by the human body. In dysbiosis, this equilibrium is upset. Dysbiosis is a general term used to describe a gut environment where either the quantity of beneficial organisms is significantly reduced, or the quantity of harmful organisms is significantly increased, or the overall diversity of organisms is reduced [27]. Dysbiosis is associated with various bowel-related diseases, including ulcerative colitis, Crohn’s disease, inflammatory bowel disease, obesity, central nervous system disorders [28], and recently, even PJI [29]. The link between PJI and gut microbiota dysbiosis (GMD) has recently been researched. One of the main theories that suggests a potential relationship between PJI and GMD is the “Trojan horse theory”. It states that organisms from distant sites may be transported from one site to another by leukocytic phagocytosis [30]. In PJI, the ingested microbes might be released in the vulnerable periprosthetic region and cause infection. Other dysbiosis-related factors that may have a link with PJI include increased gut permeability (due to the loss of integrity of the GI barrier), inflammatory bowel diseases, and the ingestion of antibiotics. 

The objective of this review is to analyse the links between the GM and PJI when compared to healthy controls. Specifically, we look at the PJI risk and microbiota profile of patients/murines with GMD compared to healthy controls to examine any potential connections. The totality of the human and mouse/rat studies are collected and analysed to create an up-to-date review and to identify any existing gaps in research, suggesting potential fields for future studies. Where possible, we use the available data to suggest how this knowledge may be utilised by health professionals to better prevent, identify, and treat PJI caused by GMD. The two questions we aim to answer are: (1) Does GMD increase the risk of PJI in murines when compared to healthy individuals?, and (2) Does GMD increase the risk of periprosthetic infection in adults when compared to healthy individuals?

## 2. Materials and Methods

### 2.1. Eligibility Criteria and Search Strategy

All the authors initially agreed on the objectives of the study, determining the main methodological strategies and the inclusion and exclusion criteria of the study. The major medical databases, PubMed, MEDLINE, and Cochrane Database of Systematic Reviews were analysed on 13 March 2023, using the following keywords: periprosthetic AND infection AND (microbiota OR flora OR microflora OR microbiome OR microbial OR dysbiosis) AND (gut OR intestinal OR enteric), searching for relevant publications on gut dysbiosis and PJI. The databases were filtered for studies published between January 1980 and March 2023, in English language.

The study selection process is summarized in Figure 1. The initial search yielded 155 papers, and after the duplicate removal 125 papers were eligible for retrieval. The titles and abstracts of the identified papers were analysed and screened independently by two authors (KV, LT). Based on the title and abstract, 110 studies were excluded because they were considered not relevant to the present study. The remaining 15 papers were evaluated in detail to verify their congruence with the inclusion criteria. Inclusion criteria were studies describing the animal and murine in English. Conversely, articles regarding aged or infant/young populations, in other languages, case reports, other animals, unpublished studies were considered as exclusion criteria. If there was any doubt about whether an article should be included, the senior authors (CF, ADM) made the final choice. References of relevant papers were then analysed to find additional works pertinent to the topic. After full text screening, a total of 5 studies were selected meeting the inclusion criteria.

### 2.2. Risk of Bias Assessment

The Newcastle–Ottawa risk of bias tool was used to evaluate cohort studies. The total score out of nine gives an indication of the bias present in the study with a higher score suggesting a lesser risk of bias. The SYRCLE risk of bias tool was used to evaluate animal studies. Similarly, the total score out of 9 indicates the risk of bias. It should be noted that just because a risk of bias was not explicitly addressed in the study does not necessarily mean that it was not accounted for by the researchers. The evaluation was performed independently by two researchers (MB and CDA). Where results were not in agreement, they were discussed with senior authors (ADM, FT) to reach a mutual conclusion.

## 3. Results

### 3.1. Review of the Included Studies

The database search retrieved five studies, whose findings and characteristics are summarized in Table 1. These included three human studies and two murine studies: one rat study and one mouse study.

In 2019, Hernandez et al. conducted a mouse study to test the effect of an antibiotic-altered gut microbiota on the risk of developing PJI [35]. Eighty-six four-month-old C57Bl/6 mice were divided into two groups, one being administered neomycin and ampicillin (antibiotics with poor oral bioavailability to limit the effects to the GI microbiome) and one left untreated. Antibiotics were administered orally by being dissolved in the water supply of the study group at 1.0 g/L for ampicillin and 0.5 g/L for neomycin. After three months of administration, both groups underwent a titanium tibial implant to simulate a joint implant. Mice were also locally inoculated with *S. aureus* in the synovial space of the implant site. Microbiota analysis was performed by faecal sample analysis. After 5 days, mice were euthanised and the implant was removed for CFU count analysis. The bacterial load in the surrounding joint tissues was greater in animals with an established infection than animals that were uninfected. Additionally, the study group had a significantly higher PJI incidence compared to the control group. In 2020, Zhu et al. conducted an animal study on rats to test the Trojan horse theory [34]. Sixty healthy adult rats were intestinally colonised with MRSA and had a K-wire inserted in their knee joints either at 8 or 72 h after MRSA colonisation (*n* = 20 each). The K-wire was used to simulate a knee arthroplasty. Mice were euthanised at 10 days and tissues were collected for analysis of PJI. Continuous intestinal MRSA presence in all mice was confirmed by frequent stool analysis. Each group consisted of 20 mice. In the 8 h group, 25% of rats developed PJI, while in the 72 h group, 10% of rats did. The experiment was repeated with MRSA colonisation using isolates taken from PJI patients, where 30% of rats in the isolate 1 group developed PJI and 35% in the isolate 2 group did. As a control for the above, a group of mice had the K-wire inserted without colonisation. None of the mice in this control group developed PJI.

In 2022, Remily et al. performed a large cohort study analysing US patients who underwent primary total knee arthroplasties between 2010 and 2020 [31]. The patients were split into three cohorts: control, Crohn’s disease, and ulcerative colitis. The odds ratio for PJI in the two inflammatory bowel disease (IBD) groups was calculated using multivariable logistic regression and adjusted for obesity, tobacco usage, and diabetes (all three had significantly higher incidence in the IBD groups). The control group consisted of 1,360,820 patients, with a Crohn’s disease group of 8369 and an ulcerative colitis group of 11,347 patients. The adjusted risk odds ratio for PJI at 90 days post-operation was 1.37 for the Crohn’s disease group and 1.31 for the ulcerative colitis group, both of which were significantly higher than the controls. In 2022, Chisari et al. conducted a retrospective cohort study of 608 patients who underwent primary hip/knee arthroplasty for osteoarthritis between 2000 and 2018 [33]. In total, 152 patients affected by IBD (either Crohn’s disease, ulcerative colitis, or regional nonspecific colitis) were matched 1:3 to control patients (*n* = 456). The follow-up period was 2 years. IBD patients were significantly more likely to develop PJI compared to controls. Using univariable cox regression analysis, IBD was found to be an independent factor for the development of PJI, and it was associated with a 5.4-fold higher risk of developing PJI in the 2-year time span (*p* = 0.007). In the same year, Chisari et al. conducted a prospective cohort study of 134 patients about to undergo primary or revision knee or hip arthroplasty. They were evaluated by intravenous blood collection for their Zonulin, sCD14, and LPS levels (markers of gut permeability/inflammation) [32]. These patients were split into four arms for analysis, according to the type and reason for arthroplasty. In total, 26 of these patients underwent primary arthroplasty at the hip/knee (arm 1). Of the revision surgery patients, 14 were classified as undergoing revision for acute PJI (<90 days post-operation, arm 2), 30 for chronic PJI (>90 days post-operation, arm 3), and the other 90 for aseptic loosening (arm 4). sCD14 and Zonulin were found to be significantly higher in PJI patients vs. non-PJI patients. Zonulin was also found to be significantly higher in acute PJI cases compared to chronic PJI cases (*p* = 0.005).

### 3.2. Study Quality Assessment and Risk of Bias Assessment

Manuscripts were published between 1980 and 2023. According to the Newcastle–Ottawa evaluating score system for cohort studies, one manuscript reported 7/16 points [31], and two reported 8/16 points [32,33]; the SYRCLE evaluating score system for animal studies showed that one study achieved 2/18 points [35], and one achieved 3/18 points [34]. The animal studies presented showed limitations in their methodology. These were observational studies of populations without rigid selection criteria and had major shortcomings both in performance and detection, as evidenced by the SYRCLE score (Table 2). Conversely, the selected human studies presented a better methodology as highlighted in their Newcastle–Ottawa Risk scores (Table 3). To obtain a correct interpretation of the data, the different values of the articles should be considered.

## 4. Discussion

The determination and characterization of the connection between gut microbiota and PJI development risk is important to establish, as it will improve the current understanding of the physiopathology of PJI formation. Most importantly, it could improve our understanding on the formation of biofilm, the major determinant of the clinical PJI complications. The examined studies showed varying results and theories in trying to elucidate the connection between GMD and PJI. Similarly to smoking and obesity [37], GMD could become one of the risk factors checked for and managed before elective surgical procedures to decrease or assess the risk of developing PJI. The findings of the current review suggest that there is indeed a connection between GMD and PJI risk, with varying degrees of evidence.

### 4.1. Limitations

This review does have some limitations. First, the results are limited by the quality of the studies included. Both animal studies have the limitation of not including sufficient randomization and blinding; the researchers and assessors of the results were not blinded from knowing the group the murines were placed in. As can be seen in Table 2 and Table 3, not all the studies had an equally thorough protocol for moderating the risk of bias. The animal studies, in particular, scored rather low in the SYRCLE test, and therefore, these carry an increased risk of bias. Moreover, even with high scoring studies, there is always a risk of omitted/missing data. This becomes an even more pressing issue in systematic reviews such as this, in which the selection process left a limited number of studies. Additionally, all the studies were written on different types of dysbiosis; therefore, while certain conclusions may be drawn on the chance of an existing link, it is difficult to ascertain the connection for any individual type of dysbiosis.

Looking at the murine studies, Zhu et al. concluded that intestinal MRSA could cause PJI [34]. The main drawback of this study is that it was an observational cohort study, while the human studies were structured with a control group that included healthy and infected populations. The results suggested that the Trojan horse theory could be linked to gut infections. This is important as MRSA-caused PJI tends to be more severe than PJIs caused by other microorganisms [38].

Hernandez et al. conducted the other murine study in this review [35], this one on mice. The study had the benefit of testing a variable with broader applications: a link to reduced gut diversity increasing the risk of PJI. The authors found such a link to be present. However, a drawback of this study was that the mice were exposed to *S. aureus* on the surgical site to induce infection. This prohibited the Trojan horse theory from being examined as it would have been impossible to deduce if any PJI caused by *S. aureus* was from the artificial exposure to the surgical site or if it originated from the gut. Despite that, the results of the study support the theory that gut dysbiosis increases the risk of PJI. The increased risk may have, in part, occurred due to a decreased immune response as a direct result of the lowered microbiota diversity. It could also have been caused by the decreased nutrient absorption, creating downstream effects. It is uncertain if the relationship between mice PJI and dysbiosis outlined in the study corresponds to a similar relationship in humans because of the differences in their immune system. Therefore, some caution should be exercised in interpreting the results.

Analysing the human studies, the retrospective cohort study of Remily et al. had a considerably large sample size [31]. The two study groups of IBD, namely Crohn’s disease and ulcerative colitis, had a combined patient number of 19,716. The significantly increased odds ratio found in both IBD groups (*p* < 0.01) and the results are therefore quite reliable. Unfortunately, the adjusted risk ratio was only calculated for acute PJI risk (<90 days). It could be of interest to examine the odds ratio of long-term PJI risk; above all, because PJI case numbers were collected at 1- and 2-year follow-up. The study only looked at knee infections, and therefore, results could differ when other joints’ sites are considered.

Chisari et al., on the other hand, provided 1- and 2-years post-operative data regarding the risk of developing PJI in IBD and control patients [33]. Therefore, they offered some evidence on the long-term PJI risk for IBD patients, which Remily et al. did not provide. The results of the study demonstrated that, in knee and hip arthroplasty, the risk of PJI was considerably higher in IBD patients compared to non-IBD controls.

Finally, the second study by Chisari et al. examined the relationship between gut permeability and PJI [32]. The fact that sCD14 and Zonulin were significantly higher in PJI patients suggests that permeability may play a role in the risk to develop an infection. Zonulin used as a marker gives an indication of the “leakiness” of the gut. Decreased Zonulin levels may prevent the onset of rheumatoid arthritis by preventing the immigration of leukocytes to joint sites [39]. This finding is in line with the Trojan horse theory, as the leukocytes involved in arthritis could also release bacteria to the periprosthetic joint site, triggering PJI. sCD14 is a marker of inflammation/immunologic response [40]. Its increase alongside Zonulin suggests an increased leukocyte transfer from the gut to the rest of the body. The fact that PJI patients showed significantly higher gut permeability may indicate that gut dysbiosis plays a primary role in causing PJI, even in non-IBD patients (as patients suffering from IBD were excluded from the study). However, the concept of a “leaky gut” is still under evaluation, and any assumption should be carefully evaluated from the perspective of future research [41]. Moreover, available ELISA tests are not specific or sensitive enough to Zonulin, and therefore its widespread clinical application is under question [42]. New serum markers with better specificity and sensitivity should be identified to better study this pathology to simplify the diagnosis process and aid treatment.

### 4.2. Considerations

At present, the most important consideration in this trending medical field is primary prevention, which may include practices to reduce infective risk during joint replacement. Natural barriers, such as skin and nasal mucosae, can act as barriers to infection, and their decolonization before surgery from known pathogens may reduce potential sources of infection [43]. In this view, the GI tract represents another crucial barrier that, when altered by diseases targeting the GI tract (including HIV, celiac disease, and irritable bowel syndrome) may increase the risk of PJI [44]. In addition to the prevention of gut dysbiosis, it is necessary to treat and counteract those chronic diseases that may alter its GI permeability. To mitigate the impact of gut dysbiosis on PJI, probiotics and prebiotics can be used to restore or maintain a healthy gut microbiota during and after antibiotic treatment, potentially reducing the risk of complications [45].

The field has potential for a scientific breakthrough, given the positive results displayed in the data. First, more research should be performed in mouse and rat models, to establish if there is a link between PJI risk and the main types of dysbiosis. Ideally, common inflammatory causes for the GI, including Crohn’s disease and ulcerative colitis, should be examined. The types of leukocytes that are most likely to act as “Trojan horses” should be fully characterized so that future research can target prevention and treatment. Research should simulate how the risk of PJI varies in younger and geriatric murines, as their immune system differs from adults. Given that IBD patients show an increased risk of PJI, research in humans should focus on methods to mitigate the risk of infection in these populations. Researchers should analyse both the risk for acute (≤90 days) and chronic PJI (>90 days) to develop, in order to establish if the 90-day risk is significantly higher compared to the long-term risk. If this is the case, patients may benefit from surgeries scheduled in periods between IBD flare-ups, to decrease the chances of leukocytes acting as “Trojan horses” and contributing to PJIs. To analyse the Trojan horse theory, it will be important to determine the best methods for classifying and detecting leukocytes. Samples from the periprosthetic region can be collected and analysed with techniques such as single-cell sequencing. In single-cell sequencing, the samples are analysed for their RNA content, and converted to cDNA, which can then be amplified so as to find traces of existing or broken down leukocytes [46]. Through data analysis, the results can then be classified to the type of leukocyte that was present. This procedure is suggested as it is highly sensitive—the infection vector could be a single leukocyte after all.

In non-IBD patients, dysbiosis could be more easily managed, and patients may benefit from gut permeability and inflammation marker analysis pre-operation. If these are determined to be high, a consultation with a gastroenterologist may establish the cause of dysbiosis and reduce the risk of PJI before surgery is scheduled. Research should also target the full characterization and treatment of the “leaky gut”, its potential reversal by pre- and probiotics, and how such treatments affect the incidence rate of developing PJI. The concept of a “leaky gut” is not entirely understood. It is hypothesised to occur due to increased intestinal permeability caused by various factors. Factors that were found to be associated with a state of increased intestinal permeability include endurance exercise, liver diseases, esophagitis, neurological diseases, psychiatric diseases, food allergies, altered gut microbiota (as seen in this review), altered metabolism, aging, and pharmaceutical intervention aimed at the gut. Its current methods of measurement include fractional urinary excretion of orally administered probe molecules (in vivo and in vitro using a mucosal biopsy) and endoscopic measurements of the intestinal barrier. None of the current methods are, however, universally accepted, and their medical significance remains unclear [47]. Thus, future research on the measurement protocols would aid in both clarifying the role of the gut bacteriome in PJI as well as potentially finding a commonly accepted standard to quantitatively measure “leaky gut” states. The usage of prebiotics to treat IBD showed limited efficacy; however, their usage in non-IBD patients is not contraindicated [44]. Additional cohort studies could outline how different factors affect the risk of gut-induced PJI, and their role in the determination and treatment of acute and chronic PJIs.

Finally, it is important to consider the role of other microorganisms that together make up the gut microbiota. Fungi exist in large numbers in the intestine and can easily colonise other sites in the human body [48]. Fungal infections of the PJI, although rare (accounting for <2% of PJI cases), represent a medical challenge as they currently have a poor prognosis and have many treatment-resistant strains. They are believed to be a more prevalent case of PJI in cases of immunosuppression [49]. As well as fungi and bacteria, PJI has not been found to be caused by other microorganisms such as archaea, protists, and algae [50], However, these other microorganisms could still influence the risk of PJI when there is a state of dysbiosis in the gut. This can be observed in Hernandez et al. where the overall microbiota diversity being lowered was associated with an increased risk of PJI. Likewise, this is further demonstrated in the two Chisari et al. studies and the Remily et al. study, which all found a link between IBD and PJI, as archaea [51,52,53,54] have been shown to influence the overall gut microbiota health and the pathogenesis of IBD symptoms. Another influence on the gut microbiota, and subsequently, on PJI risk are intestinal bacteriophages—enteroviruses—which influence the enteric bacteriome. In IBD cases, the enteric virome has been found to be increased in quantity but decreased in diversity. The enterophages are, however, not well classified taxonomically, so it is difficult to draw conclusions on the role of the enteric virome on gut dysbiosis [55].

## 5. Conclusions

In summary, the results of the above studies suggest that gut dysbiosis does increase the risk of PJI in both murine and humans, albeit with limited evidence. PJI patients were associated with altered gut permeability and a rise in inflammation markers, suggesting an underlying dysbiosis. This is associated with an increased risk of PJI caused by decreased gut biodiversity, the presence of IBD, increased gut inflammation/permeability, or by an alteration in the healthy gut microbiota ratio.

## Figures and Tables

**Figure 1 microorganisms-11-01778-f001:**
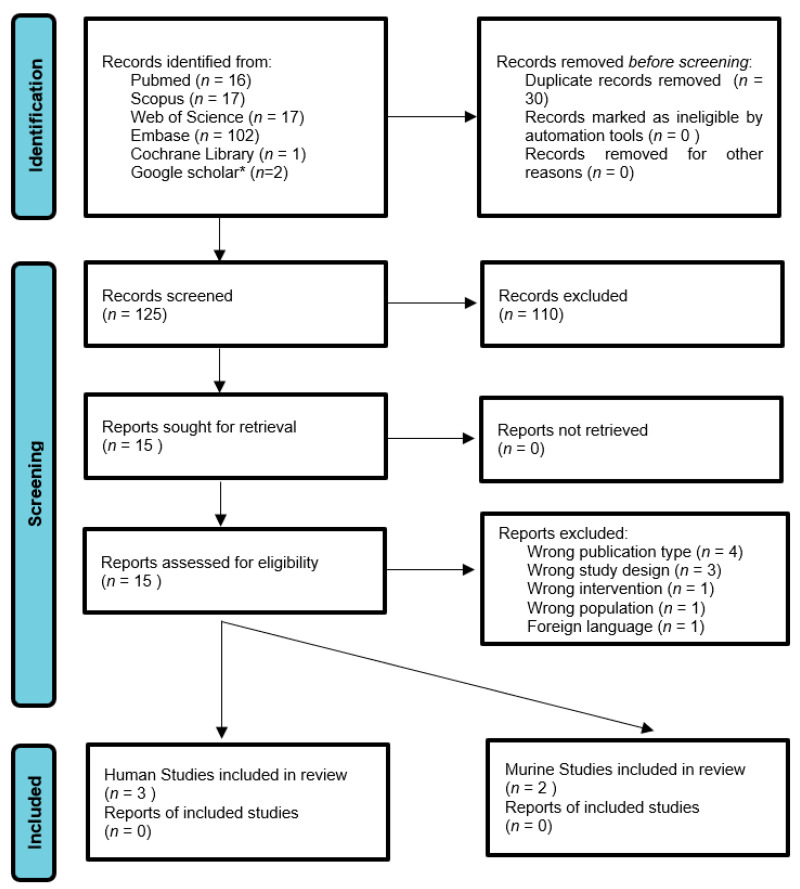
Flowchart for manuscript selection in the current study. Search was performed on 14 March 2023. * Google scholar was searched up to the first 20 pages.

**Table 1 microorganisms-11-01778-t001:** Characteristics and findings of included studies.

Authors	Date Published	Type of Study	Dysbiosis Type	N. of Patients/ Subjects	Length of Follow-Up	Analysed Variables	Results
Remily et al. [31]	12/2022	Retrospective Cohort Study	Crohn’s Disease and Ulcerative Colitis	1,380,536	2 years	PJI cases in control, Crohn’s disease, and ulcerative colitis patients. Results adjusted for confounding variables of obesity, tobacco usage, and diabetes	PJI odds ratio at 90 days: 1.37 for Crohn’s disease (*p* < 0.01) and 1.31 for ulcerative colitis (*p* < 0.01)
Chisari et al. [32]	9/2022	Prospective Cohort Study	IBD—Gut Permeability	134	n/a	Zonulin, LPS, sCD14 (markers of gut permeability)	CD14 and Zonulin higher in PJI vs. non-PJI patients (555 ± 216 ng/mL, *p* < 0.05 and 7.642 ± 6.077 ng/mL, *p* < 0.001 respectively). Zonulin higher in acute PJI group vs. chronic (10.7 ± 6.2 ng/mL vs. 5.8 ± 4.8 ng/mL, *p* = 0.005)
Chisari et al. [33]	1/2022	Retrospective Cohort Study	IBD	608	2 years	PJI cases in IBD patient group and non-IBD group. Analysed using univariable regression with a cox regression model	IBD group more likely to develop PJI vs. non-IBD control (4.61% vs. 0.88%, *p* = 0.0024). IBD associated with 5.4-fold higher incidence of PJI (*p* = 0.007)
Zhu et al. [34]	4/2020	Preclinical—Rat Study	Intestinal MRSA	100	10 days	Stool analysis on days 1–10. After sacrifice, specimens from knee joint, femur, and implanted K-wire harvested and cultured for bacterial identification. 16S ribosomal DNA, GFP gene expression, and PCR used for analysis	MRSA strain: 25% of rats in the 8 h group and 10% of rats in the 72 h group developed PJI. MRSA isolate: 30% of rats in isolate 1 and 35% in isolate 2 developed PJI. Control: 0 rats developed PJI
Hernandez et al. [35]	7/2019	Mouse study	Altered gut microbiota due to antibiotics	82	5 days	Bacterial load (CFU) on implant surface. Shannon diversity index for microbiome diversity	Higher PJI occurrence in altered gut microbiota mice vs. control (72.5% vs. 50%, *p* = 0.03) and lower microbiome diversity (2.36 ± 0.57 vs. 3.87 ± 0.28, *p* < 0.001)

**Table 2 microorganisms-11-01778-t002:** SYRCLE’s Risk of Bias tool for Animal Studies [36].

Study	Study Design	Selection			Performance		Detection		Attrition	Reporting	Total Score
		Sequence Generation	Baseline Characteristics	Allocation Concealment	Random Housing	Blinding Researchers	Random Outcome Assessment	Blinding Assessors	Incomplete Outcome Data	Selective Outcome Reporting	
Hernandez et al. [35]	Mouse Study	+	+	−	−	−	−	−	+	−	3
Zhu et al. [34]	Rat Study	−	+	−	−	−	−	−	+	−	2

**Table 3 microorganisms-11-01778-t003:** Newcastle–Ottawa Risk of Bias tool for Cohort Studies (https://www.ncbi.nlm.nih.gov/books/NBK115843/bin/appe-fm3.pdf. accessed 6 June 2023).

Study	Study Design	Selection				Comparability	Outcome			Total Score
		Exposed Cohort Representative?	Selection of Non-Exposed Cohort	Ascertainment of Exposure	Outcome Not Present at Study Start?	Based on Design or Analysis	Assessment of Outcome	Timing of Follow-Up	Adequate Follow-Up	
Chisari et al. [32]	Cohort Study	+	+	+	+	++	+	+	−	8
Chisari et al. [33]	Cohort Study	+	+	+	+	++	+	+	−	8
Remily et al. [31]	Cohort Study	−	+	+	+	++	+	+	−	7

## Data Availability

All data included in this review are publicly available in their respective cited study.

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
