# Peer review of "The Potential Role of Gut Bacteriome Dysbiosis as a Leading Cause of Periprosthetic Infection: A Comprehensive Literature Review"

_microorganisms, 2023, doi:10.3390/microorganisms11071778_

Round 1
Reviewer 1 Report
A very well-conducted review of the available literature on the potential role of intestinal dysbiosis as the main cause of periprosthetic infections. The work is well thought out and carefully written. I have no comments.
Author Response
Comment: A very well-conducted review of the available literature on the potential role of intestinal dysbiosis as the main cause of periprosthetic infections. The work is well thought out and carefully written. I have no comments.
Response to comment: We thank the Reviewer.
Reviewer 2 Report
The manuscript by Valtetsiosis et al. demonstrated a systematic review and indicated that gut dysbiosis is linked to an increased risk of Periprosthetic Joint Infections. This is an interesting topic and the manuscript was in general well organized with results that in principle support their conclusions. However, the authors are suggested to simplify the results of Table1. The authors are also suggested to add the discussion about the limitation of this reivew.
Author Response
The manuscript by Valtetsiosis et al. demonstrated a systematic review and indicated that gut dysbiosis is linked to an increased risk of Periprosthetic Joint Infections. This is an interesting topic and the manuscript was in general well organized with results that in principle support their conclusions.
Comment 1: However, the authors are suggested to simplify the results of Table 1.
Response to comment 1: We thank the Reviewer. The results of Table 1 have now been simplified.
Comment 2: The authors are also suggested to add the discussion about the limitation of this review.
Response to comment 2: We thank the Reviewer. The discussion part explaining the review’s limitations has now been modified accordingly.
Reviewer 3 Report
The manuscript entitled “The potential role of gut dysbiosis as a leading cause of periprosthetic infection: a systematic review” reviewed the current knowledge on the role of gut microbiota dysbiosis in increasing the risk of periprosthetic joint infection. The authors systematically screened 155 published papers and finally focused on 5 recently published articles investigating the relationship between gut microbiota dysbiosis and periprosthetic joint infection. This review is important to raise additional attention about the role gut microbiota. The topic of this manuscript is appropriate for this type of journal. The review was well written. However, some major and minor points are required to be improved as below:
Major revision comments:
According to paper entitled “Microbiome definition re-visited: old concepts and new challenges” (link: https://microbiomejournal.biomedcentral.com/articles/10.1186/s40168-020-00875-0). The definition of microbiota includes bacteria, archaea, fungi, protists, alage. In addition, bacteriophages also play a potential role in modulate the microbiome, which was mentioned in the paper “Phages and their potential to modulate the microbiome and immunity” (link: https://www.nature.com/articles/s41423-020-00532-4) This review focused in dysbiosis of gut bacteria instead of the whole microbiota, therefore, it would be important to narrow down the aspects to changes in gut bacteria.
In the discussion part, it would be essential if the authors can provide some opinions related to the other components of microbiota such as fungi, bacteriophage, etc in affecting the risk of periprosthetic joint infection
Line 138: Please provide the supplementary file including the information of the 125 papers which were eligible for retrieval
Line 154: Please provide the reference of the methods used for risk of bias assessment
Minor revision comments:
Title: “gut dysbiosis” describes a quite general concept. In this review, the authors mainly concentrate on the role of gut bacteriome dysbiosis. I suggest the authors to consider more specific words
Line 25: do the authors mean Crohn's disease?
Line 135: Please change "periprosthetic joint infection" to "PJI"
Line 175: please change "S. aureus" to italicized font
Line 182: please check whether it is "MRA" or "MRSA"
Line 325: Please provide your opinions of which methods can be used to investigate and classify the leukocytes acted as Trojan horses". One cutting-edge technique can be considered is single-cell sequencing, it would be nice if the authors can provide more information about that
Line 326: Please change "trojan horses" to "Trojan horses"
Line 331: As the author mentioned in lines 296-297. The concept of "leaky gut" is under investigation. The definition and standard methods in determining a leaky gut are unclear. However, the paper entitled “The Leaky Gut: Mechanisms, Measurement and Clinical Implications in Humans” (link: https://www.ncbi.nlm.nih.gov/pmc/articles/PMC6790068/) described some mechanisms and measurement methods. It would be important if the authors could consider in this review.
Line 340-341: Please check the grammar of the sentence "it be due to decreased gut diversity"
Line 354-454: Please check the reference format. Some co-authors’ names were not listed in the references
Please check minor English errors as mentioned in the minor revision comments
Author Response
Comment 1: According to paper entitled “Microbiome definition re-visited: old concepts and new challenges” (link: https://microbiomejournal.biomedcentral.com/articles/10.1186/s40168-020-00875-0). The definition of microbiota includes bacteria, archaea, fungi, protists, alage. In addition, bacteriophages also play a potential role in modulate the microbiome, which was mentioned in the paper “Phages and their potential to modulate the microbiome and immunity” (link: https://www.nature.com/articles/s41423-020-00532-4) This review focused in dysbiosis of gut bacteria instead of the whole microbiota, therefore, it would be important to narrow down the aspects to changes in gut bacteria.
Response to comment 1: We thank the Reviewer for the clarification. As has been suggested, we have now altered the wording throughout the text to reflect the fact that the paper is focused on gut bacteria and not the microbiota in general. We have also added an explanation on the introduction “The gut microbiota is composed of a diverse range of organisms, including bacteria, archea, fungi, protists, and algae” (line 85-87) and “Bacteriophages and the gut virome they create are also considered to have an important role in modulating the gut microbiota” (line 102-104) with the relevant references.
Comment 2: In the discussion part, it would be essential if the authors can provide some opinions related to the other components of microbiota such as fungi, bacteriophage, etc in affecting the risk of periprosthetic joint infection
Response to comment 2: We thank the Reviewer. We have now added in the discussion: “Finally, it is important to consider the role of other microorganisms that together make up the gut microbiota. Fungi exist in large numbers in the intestine and can easily colonise other sites in the human body (https://www.sciencedirect.com/science/article/pii/S1438422121000199?via%3Dihub). Fungal infections of the PJI, although rare (accounting for <2% of PJI cases), represent a medical challenge as they currently have a poor prognosis and have many treatment-resistant strains. They are believed to be a more prevalent case of PJI in cases of immunosuppression (https://www.sciencedirect.com/science/article/pii/S2001037023001204). Besides fungi and bacteria, PJI is not found to be caused by other microorganisms such as archaea, protists, and algae (https://www.ncbi.nlm.nih.gov/pmc/articles/PMC6572185/ , https://sci-hub.wf/10.1016/j.cmi.2021.06.006). However, these other microorganism could still influence the risk of PJI when there are in a state of dysbiosis in the gut. This may be observed in Hernandez et al. where the overall microbiota diversity being lowered was associated with an increased risk of PJI. Likewise, this is further demonstrated in the two Chisari et al. studies and the Remily et al. study which found a linked between IBD and PJI, as archea (https://www.frontiersin.org/articles/10.3389/fphys.2021.783295/full , https://www.ncbi.nlm.nih.gov/pmc/articles/PMC8468012/), protists (https://www.ncbi.nlm.nih.gov/pmc/articles/PMC9251125/, https://www.ncbi.nlm.nih.gov/pmc/articles/PMC5680167/) have been shown to influence the overall gut microbiota health and the pathogenesis IBD symptoms. Another influence on the gut microbiota, and subsequently on PJI risk are intestinal bacteriophages – enteroviruses – which influence the enteric bacteriome. In IBD cases the enteric virome has been found to be increased in quantity but decreased in diversity. The enterophages are however not well classified taxonomically so it is difficult to draw conclusions on the role of the enteric virome on gut dysbiosis.” (Lines 517-545)
Comment 3: Line 138: Please provide the supplementary file including the information of the 125 papers which were eligible for retrieval.
Response to comment 3: We thank the Reviewer. The supplementary file with the information of the 125 papers will be included in the resubmission of the article.
Comment 4: Line 154: Please provide the reference of the methods used for risk of bias assessment
Response to comment 4: We thank the Reviewer. We have now added references for the two risk of bias tools we used, please see Table 2 and 3.
Minor revision comments:
Comment 1: Title: “gut dysbiosis” describes a quite general concept. In this review, the authors mainly concentrate on the role of gut bacteriome dysbiosis. I suggest the authors to consider more specific words
Response to comment 1: We thank the Reviewer. The title has been updated to “The potential role of gut bacteriome dysbiosis as a leading cause of periprosthetic infection : a comprehensive literature review”.
Comment 2: Line 25: do the authors mean Crohn's disease?
Response to comment 2: We thank the Reviewer. The word has been corrected to Crohn’s disease.
Comment 3: Line 135: Please change "periprosthetic joint infection" to "PJI"
Response to comment 3: We thank the Reviewer. The line has been changed accordingly.
Comment 4: Line 175: please change "S. aureus" to italicized font
Response to comment 4: We thank the Reviewer. S. aureus has now been italicized here and all its other appearances.
Comment 5: Line 182: please check whether it is "MRA" or "MRSA"
Response to comment 5: We thank the Reviewer. The word has now been changed to MRSA.
Comment 6: Line 325: Please provide your opinions of which methods can be used to investigate and classify the leukocytes acted as Trojan horses". One cutting-edge technique can be considered is single-cell sequencing, it would be nice if the authors can provide more information about that
Response to comment 6: We thank the Reviewer. Single-cell sequencing has now been included as suggested as “To analyse the Trojan horse theory, it would be important to determine the best methods in classifying and detecting leukocytes. Samples from the periprosthetic region would be collected and analysed with techniques such as single cell sequencing. In single-cell sequencing the samples would be analysed for their RNA content, converted to cDNA, which would then be amplified so as to find traces of existing on broken down leukocytes (https://www.ncbi.nlm.nih.gov/pmc/articles/PMC8964935/). Through data analysis, the results can then be classified to the type of leukocyte that was present. This procedure is suggested as it is highly sensitive – the infection vector could be a single leukocyte after all.” (Line 483-493).
Comment 7:Line 326: Please change "trojan horses" to "Trojan horses"
Response to comment 7: We thank the Reviewer. The word has been changed accordingly.
Comment 8: Line 331: As the author mentioned in lines 296-297. The concept of "leaky gut" is under investigation. The definition and standard methods in determining a leaky gut are unclear. However, the paper entitled “The Leaky Gut: Mechanisms, Measurement and Clinical Implications in Humans” (link: https://www.ncbi.nlm.nih.gov/pmc/articles/PMC6790068/) described some mechanisms and measurement methods. It would be important if the authors could consider in this review.
Response to comment 8: We thank the Reviewer. We have now added the following in lines 500-513 “The concept of a “leaky gut” is not entirely understood. It is hypothesised to occur due to increased intestinal permeability caused by various factors. Factors that were found to be associated with a state of increased intestinal permeability include endurance exercise, liver diseases, esophagitis , neurological diseases, psychiatric diseases, food allergies, altered gut microbiota (as seen in this review), altered metabolism, aging, and pharmaceutical intervention aimed at the gut. Its current methods of measurement include fractional urinary excretion of orally administered probe molecules (in vivo and in vitro using a mucosal biopsy) and endoscopic measurements of the intestinal barrier. None of the current methods are however universally accepted and their medical significance remains unclear (https://www.ncbi.nlm.nih.gov/pmc/articles/PMC6790068/). Thus, future research on the measurement protocols would aid in both in clarifying the role of the gut bacteriome in PJI as well as potentially find a commonly accepted standard to quantitatively measure “leaky gut” states.”
Comment 9: Line 340-341: Please check the grammar of the sentence "it be due to decreased gut diversity"
Response to comment 9: We thank the Reviewer. The sentence has been updated to “This is associated with an increased risk of PJI caused by decreased gut biodiversity, the presence of IBD, increaed gut inflammation/permeability, or by an alteration of the healthy gut microbiota ratio”.
Comment 10: Line 354-454: Please check the reference format. Some co-authors’ names were not listed in the references
Response to comment 10: We thank the Reviewer. The co-authors’ names were now added were previously omitted in the reference section.
Round 2
Reviewer 3 Report
The manuscript quality has been improved. The authors have responded to all of my comments.
The minor English typos were corrected in the updated version of the manuscript